

# Inventory statistics meet big data: complications for estimating numbers of species

Ali Khalighifar[1,2], Laura Jiménez[1,2], Claudia Nuñez-Penichet[1,2], Benedictus Freeman[1,2], Kate Ingenloff[1,2], Daniel Jiménez-García[1,3] and Town Peterson[1,2]

[1] Biodiversity Institute, University of Kansas, Lawrence, KS, USA
[2] Department of Ecology and Evolutionary Biology, University of Kansas, Lawrence, KS, USA
[3] Centro de Agroecología y Ambiente, Benemerita Universidad Autónoma de Puebla, Puebla, Puebla, Mexico

## ABSTRACT

We point out complications inherent in biodiversity inventory metrics when applied to large-scale datasets. The number of units of inventory effort (e.g., days of inventory effort) in which a species is detected saturates, such that crucial numbers of detections of rare species approach zero. Any rare errors can then come to dominate species richness estimates, creating upward biases in estimates of species numbers. We document the problem via simulations of sampling from virtual biotas, illustrate its potential using a large empirical dataset (bird records from Cape May, NJ, USA), and outline the circumstances under which these problems may be expected to emerge.

## INTRODUCTION

Biodiversity measurements have important implications for conservation efforts (*Sousa-Baena, Garcia & Peterson, 2014*). Biodiversity metrics provide information about community composition, numbers of species, and similarity or dissimilarity of species composition among sites (*Colwell & Coddington, 1994*), and can allow researchers to separate well-inventoried sites from partially-inventoried sites for macroecological analyses (*Lobo et al., 2018*). Biodiversity inventories have been implemented at scales ranging from local to global (*Moreno & Halffter, 2000*; *Ballesteros-Mejia et al., 2013*), to evaluate and understand biotic responses to changing environmental conditions.

Tracking species richness in biodiversity inventories was originally achieved via visual assessment of asymptotic behavior of species accumulation curves (*Karr, 1980*), and then with the quantitative assist of non-linear regressions (*Clench, 1979*; *Soberón & Llorente, 1993*). However, for the past 20+ years, non-parametric estimators of numbers of species have been used to estimate species richness, particularly a set of estimators based on sampling theory (*Chao, 1987*). Diverse data origins and variable data quality pose significant challenges for such analyses, particularly when data are drawn from publicly

Corresponding author
Ali Khalighifar, a.khalighifar@ku.edu

accessible databases, rather than collected individually by the researcher (*Soberón, Llorente & Benitez, 1996*; *Lobo, 2008*).

However, those same publicly accessible databases offer exciting opportunities for novel analyses (e.g., *Cameron et al., 2018*; *Peterson et al., 2015*). Primary biodiversity data connect a particular species with a place and a point in time (*Sullivan et al., 2014*), and availability of such data records has grown massively, now exceeding $10^9$ records (e.g., Global Biodiversity Information Facility, http://www.gbif.org, serving 1,387,995,196 records as of 22 January 2020). Although these data are heavily biased in terms of their spatial and temporal distributions, being concentrated massively in Europe and North America and a few other, scattered regions (*Yesson et al., 2007*; *Peterson & Soberón, 2018*), the promise of genuine, macroscale, synthetic insights remains, and is growing.

In this contribution, we report on a complication in application of the customary statistics for measuring species richness (*Colwell & Coddington, 1994*) to very large-scale (e.g., $10^6$ records or larger) biodiversity incidence datasets (i.e., records only of presence, and not of abundance). Biodiversity datasets have long been of modest dimensions only, and the field has been built on metrics and methods equipped and designed for those dimensions. In the course of studies of avifaunal change over recent decades in North America that are pending publication, we noted that species richness estimates are affected significantly by what would seem to be negligible numbers of errors among the real data records (see Fig. 1, for an example from a site that is sampled massively by birdwatchers). We present a brief conceptual summary and a demonstration of the problem via a simple simulation; we conclude with an exploration of how such problems can be avoided or mitigated.

## Conceptual background

The problem of estimating species richness from samples has been approached via methods that can be separated into three groups according to the statistical approach used to derive a species richness estimator: (1) extrapolating species accumulation curves to their asymptotes (*Clench, 1979*), (2) fitting parametric distributions of relative abundances (*Efron & Thisted, 1976*), or (3) using nonparametric techniques based on distribution of individuals among species (or the distribution of species among samples) (*Colwell & Coddington, 1994*; *Colwell, 2013*; *Chao & Chiu, 2016*). We focus on asymptotic versions of these methods sensu *Chao & Chiu (2016)*, as we are interested in full inventories of species present at sites; see discussion in *Peterson & Slade (1998)*. Two kinds of data are used in these richness studies: incidence data, in which only presences and absences are recorded for each species in each unit of effort, and abundance data, in which numbers of individuals of each species are recorded within each unit of effort (*Gotelli & Colwell, 2011*). Abundance data can always be converted to incidence data, whereas the reverse is not generally possible.

The nonparametric approach has been preferred greatly, since it does not make assumptions about underlying distributions of abundances or detection rates of species (*Chao & Shen, 2004*; *Chao & Chiu, 2016*). We focus on four nonparametric species

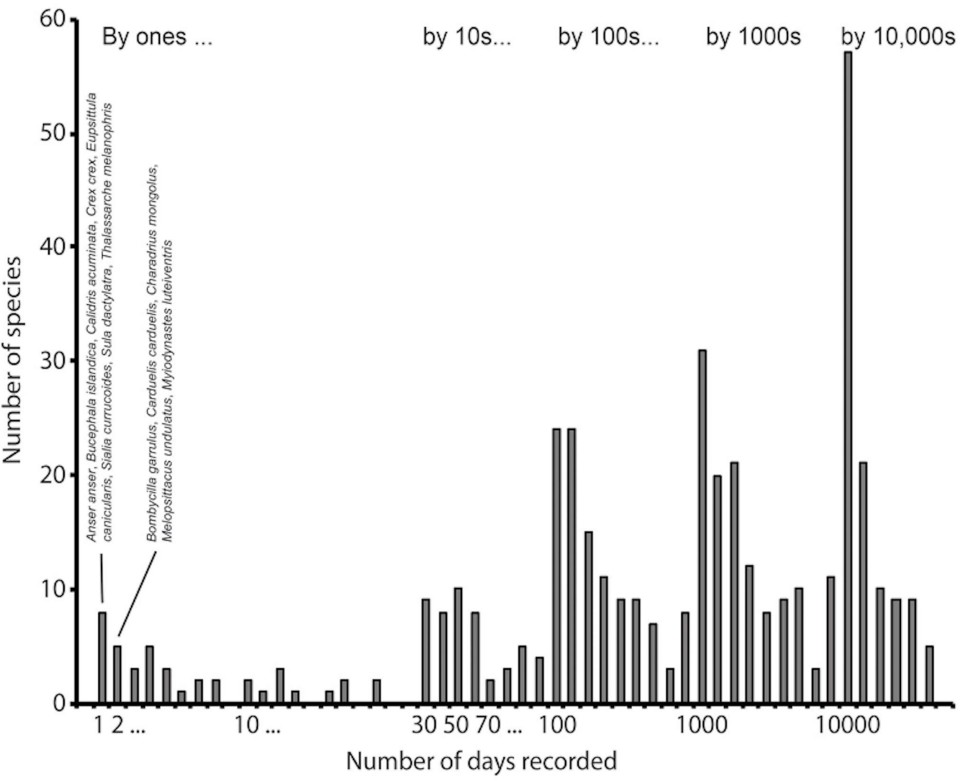

**Figure 1 Example of an intensively sampled site, Cape May National Wildlife Refuge, NJ, USA.** This example shows how the frequency histogram of number of detections per species reflects large numbers of observations of a finite biota. This histogram summarizes 12,144,561 records for the site, and 436 species detected. We have identified the species having the lowest frequencies of detection, among which can be noted several species that are probably not occurring there naturally, such as *Anser anser*, *Eupsittula canicularis*, and *Melopsittacus undulatus*, all of which are likely there as escapes from captivity.

richness estimators based on replicated incidence data that estimate numbers of species actually present at a site but not observed in the reference sample. All of the estimators correct observed richness (which is by default a lower bound for a species richness estimator) by adding a term estimating the number of species present but not detected based on numbers of species represented in one sample (uniques), two samples (duplicates), or a few samples only (*Gotelli & Colwell, 2011*; *Colwell et al., 2012*).

The reference sample for replicated incidence data consists of a species-by-effort unit (frequently these units of effort will be days of inventory effort) matrix in which each element ($m_{ij}$) corresponds to either the presence or absence of species $i$ in sample $j$. The number of columns in this matrix, $T$, is the number of effort units in the sample; the number of rows is the observed number of species, $S_{obs}$. $Q_k$ is the number of species present in exactly $k$ effort units of the sample, so the number of species present in the assemblage but not included in the overall sample (undetected species) is $Q_0$, the number of species unique to a single effort unit is $Q_1$, the number of duplicates is $Q_2$, and so on.

*Chao (1984)* originally derived an estimator of species richness $S_{obs}$ for abundance-based data that is now called *Chao1*, which she later recast for incidence data (*Chao, 1987*). This latter estimator, now called *Chao2*, is

$$\hat{S}_{Chao2} = \begin{cases} S_{obs} + \dfrac{\left[\dfrac{T-1}{T}\right]Q_1^2}{2Q_2}, & \text{if } Q_2 > 0 \\ S_{obs} + \left[\dfrac{T-1}{T}\right]\dfrac{Q_1(Q_1-1)}{2(Q_2+1)}, & \text{if } Q_2 = 0 \end{cases} \tag{1}$$

where $T$ is the sample size available for the overall calculation. The first expression of Eq. (1) reflects the classic *Chao2* estimator; however, this estimator is undefined when $Q_2 = 0$. The second expression in Eq. (1) is a corrected form that is always obtainable and defined.

A second estimator of interest, the incidence coverage-based estimator (ICE), is based on the concept of sample coverage: the proportion of the total number of incidences in a set of sampling units that belong to the species represented in the sample. Sample coverage is a measure of the information available regarding occurrence of relatively rare species in the sample (*Chao & Chiu, 2016*); its estimator depends on the complement of the proportion of singletons, in relation to the total number of incidences of the infrequent species (*Colwell & Coddington, 1994*). A third type of species richness estimator is based on the statistical method of jackknifing, a bias reduction technique involving removing subsets of the data and recalculating the estimator with the reduced sample (*Chao & Chiu, 2016*). Finally, we explored the method developed by *Chiu & Chao (2016)* for microbial molecular diversity data to account for inflation of numbers of singletons by sequencing errors (akin to identification errors); this method estimates the true value of $Q_1$ based on $Q_2$, $Q_3$ and $Q_4$, and uses the adjusted value in asymptotic diversity estimates. It is important to notice that this method defaults to the classic Chao2 estimator when both $Q_3$, and $Q_4$ are equal to zero, otherwise the estimator of $Q_1$ (the true number of uniques) would be undefined. Therefore, its application is only valid for a certain window of conditions.

Note that, for each of the estimators described above, the estimator does not take advantage of the full frequency distribution of detections for species in an inventory effort—indeed, this partial use of the frequency distribution is the focus of this contribution. Three of these estimators, as well as their corresponding variances and confidence intervals, can be computed using EstimateS (*Colwell & Elsensohn, 2014*) and a new version implemented in R (*Chao & Chiu, 2016*); the final estimator can be computed using the R version only. We used EstimateS (version 9.1.0; *Colwell & Elsensohn, 2014*) for the older three nonparametric estimators, as that platform is that which has seen the greatest use by the biodiversity community, and the R version for the latter estimator.

## MATERIALS AND METHODS

We developed a simple simulation based on large samples from a virtual community of 100 "real" species, by using a Poisson mixed model to generate the observed abundances of an inventory. Mean abundances (i.e., the parameters of the Poisson distributions) of each species were selected from a log-normal distribution with parameters $\mu = 1.5$ and
σ = 2.0 (the mean and standard deviation of the variable's natural logarithm, respectively). For each effort unit (hereafter day, for simplicity) sampled, we simulated abundances from the corresponding Poisson distributions and converted them to incidence data; that is, if an abundance was larger or equal than one, the corresponding cell in the table was set equal to 1, and it was left equal to 0 otherwise. This initial simulation served to illustrate how crucial values ($Q_1$, $Q_2$, etc.) approach zero as the frequency distribution of detections of species shifts to higher frequencies of observation, and saturates beyond the few detections on which the inventory estimators focus.

Then, to simulate effects of very rare errors in the form of misidentifications or incorrect geographic coordinates on inventory results for sites, in a second phase of simulation, we added 10 "error" species that were designed to mimic occasional, rare errors; this latter set of species had a mean abundance 6 orders of magnitude lower than the mean abundance of the 100 real species. To understand the sensitivity of our results to distributional assumptions, we also explored log-normal distributions with parameters μ = 0.3 and σ = 1.2, and μ = 1.0 and σ = 0.5, as well as gamma distributions with parameters α = 1.8 and β = 1.0, α = 2.5 and β = 2.0, and α = 3.0 and β = 1.2 (where α is the shape parameter, and β is the scale parameter).

We simulated a set of 100 replicate incidence tables of the 100 real species in R version 3.2.3 (https://www.r-project.org/) (R Core Team, 2015) and a second set of 100 replicate incidence tables of the 100 real species together with the 10 error species. To avoid recycling samples and consequent serial dependency among samples, we created independent random datasets for each number of effort units (5, 7, 10, 15, 20, 25, 50, 75, 100, 125, 150, 175, 200, 300, 400, 500, 600, 700, 800, 900, and 1,000 days), the R code can be consulted at https://github.com/LauraJim/SpeciesRichness. We used default settings of EstimateS (Colwell & Elsensohn, 2014) to calculate the Chao2, ICE, jackknife1, and jackknife2 estimators for the 100 replicates x 21 numbers of effort units = 2100 simulated populations. Next, we used customized scripts in Python 2.7.11 (the code for these analyses is available at http://hdl.handle.net/1808/25686) to separate individual replicate datasets from the combined EstimateS output files, and to select and isolate the final lines from each replicate, to create a final table of results from each simulated population. Program code for the analyses of the simulated data is available at http://hdl.handle.net/1808/25686.

## RESULTS

The results from the error-free simulations showed clearly that the estimators converged well on the true value (100 species), and that $Q_1$ and $Q_2$ approached zero in increasingly large samples (Fig. 2). The effects of adding the very rare "error" species were also quite clear: early samples lacked error species entirely, as they were just too rare to show up in relatively small samples. Only in very large samples, with 400–1,000 effort units, did these species begin to appear in the analysis datasets (red bars in Fig. 2).

The results of the simulation overall showed that, with ~150 units of effort, estimates of numbers of species in the community settled at 100 species, which is the correct number of species (Fig. 3, top). However, when rare species were introduced at minuscule abundances compared to the "real" species, even though the results settled initially on the

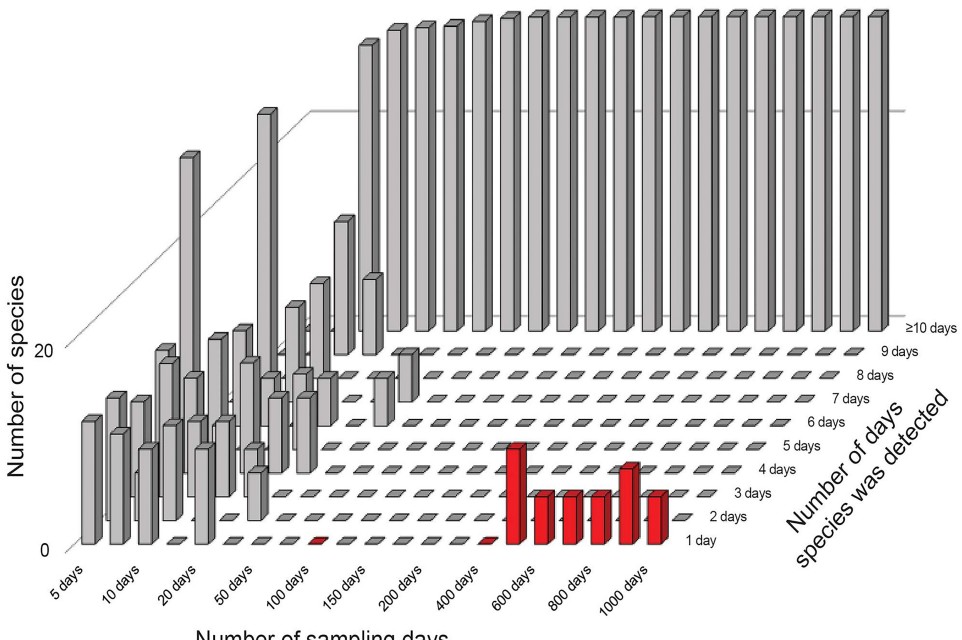

**Figure 2 Summary of frequencies of species in inventory samples used in simulation exercises.** The great bulk of these samples had large numbers of detections (the tall bars along the left and back of the figure). Note that by 50–100 days of sampling, no species are left in the 1–2 detections categories that feed into the Chao2 estimator analyses. Note also the appearance of rare species in the analysis (red bars at front right) when samples became very large.               

correct answer of 100 species, later a consistent upward bias was noted when the rare species begin to appear (Fig. 3, bottom).

The *Chiu & Chao (2016)* method showed consistent underestimation of true species numbers for modest numbers of days of sampling (Fig. 4), although this bias disappeared with large sample sizes. At modest sampling levels, although analyses of the simulated data with error better approximated the true number of species (100; Fig. 4), the consistent underestimation in error-free analyses suggests that this outcome may represent a balance between downward bias in error-free estimates and upward bias introduced by the errors.

The remaining estimators showed behavior similar to that of Chao2: ICE, Jackknife1 (first-order), and Jackknife2 (second-order) analyses, in the first simulation phase, settled on 100 species at ~100 samples, but in the second phase were biased upwards markedly by 150–250 samples (see Supporting Information). Finally, we explored different abundance distributions for the simulation—indeed, in all log-normal and gamma distributions that we assessed, biases were clear, just as in the results we have presented above (see Supporting Information).

## DISCUSSION

This contribution centers on how inventory completeness statistics need to evolve in the face of larger and larger magnitudes of biodiversity data sets. That is, we have shown that any errors in the data (e.g., misidentifications, misspellings), even at very minor frequencies, can easily end up dominating the estimation process with the common and

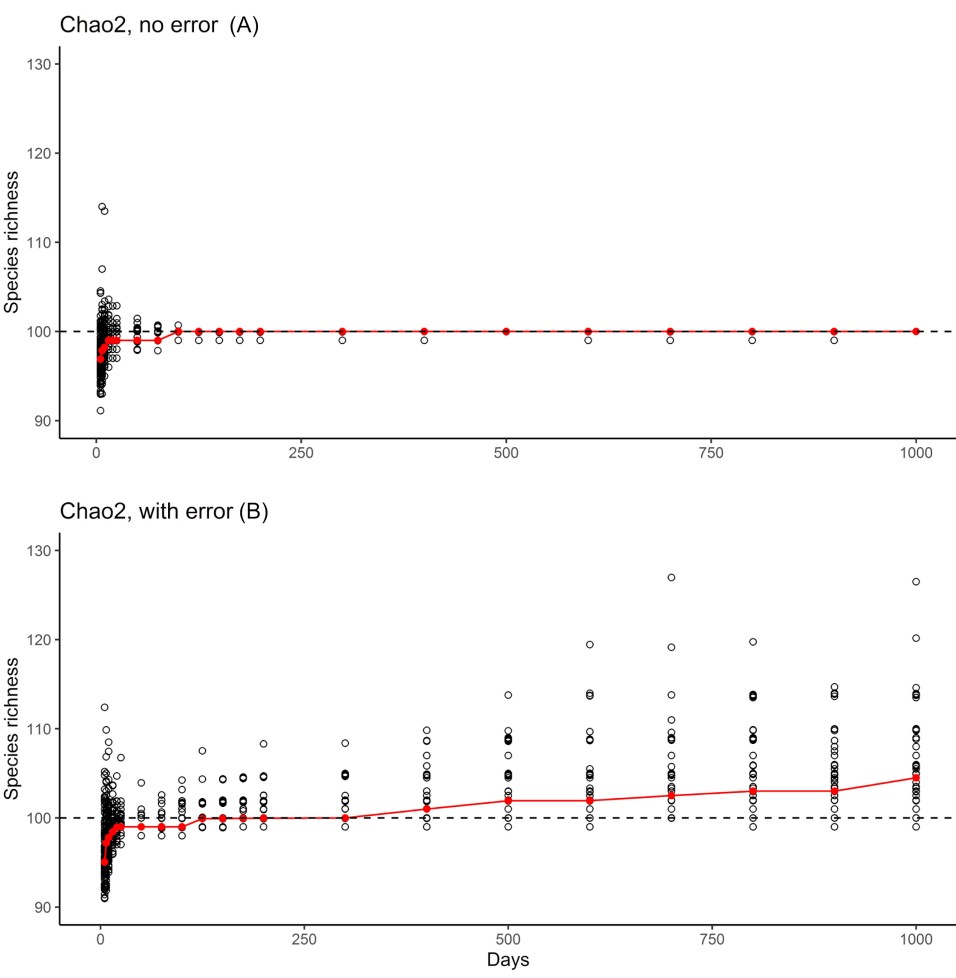

**Figure 3 Summary of the two phases of simulation results.** Graphics show simulation of accumulation of species in simulated inventory, showing the scatter of individual (black circles) and the median of results (red line). (A) One hundred real species, with no error species included. (B) One hundred real species, with 10 rare species included to simulate errors in identification or geographic references.

long-used nonparametric estimators, such as Chao2; the older species accumulation curve approach also would clearly overestimate numbers, given that "error" species would appear as species documented in the inventory. These biodiversity inventory statistics are important, offering crucial additional information to the process of biotic inventories; therefore, updating and amending these approaches to approaches that are less vulnerable to bias, or at least being cognizant of the potential for problems in estimation for big(ger) datasets, is important.

What solutions are available to a researcher with a big data set and the desire to develop detailed analyses of species richness and inventory completeness? Quite simply, a diversity of types of errors is found in pretty much every large-scale biodiversity dataset (*Lamb et al., 2009*), and large-scale datasets (see, e.g., Fig. 1) will by nature have more such errors, at least on an absolute scale. A crucial first step is that of reducing spurious and erroneous species names in the dataset (*Chapman, 2005*). Such names may be misspellings, which

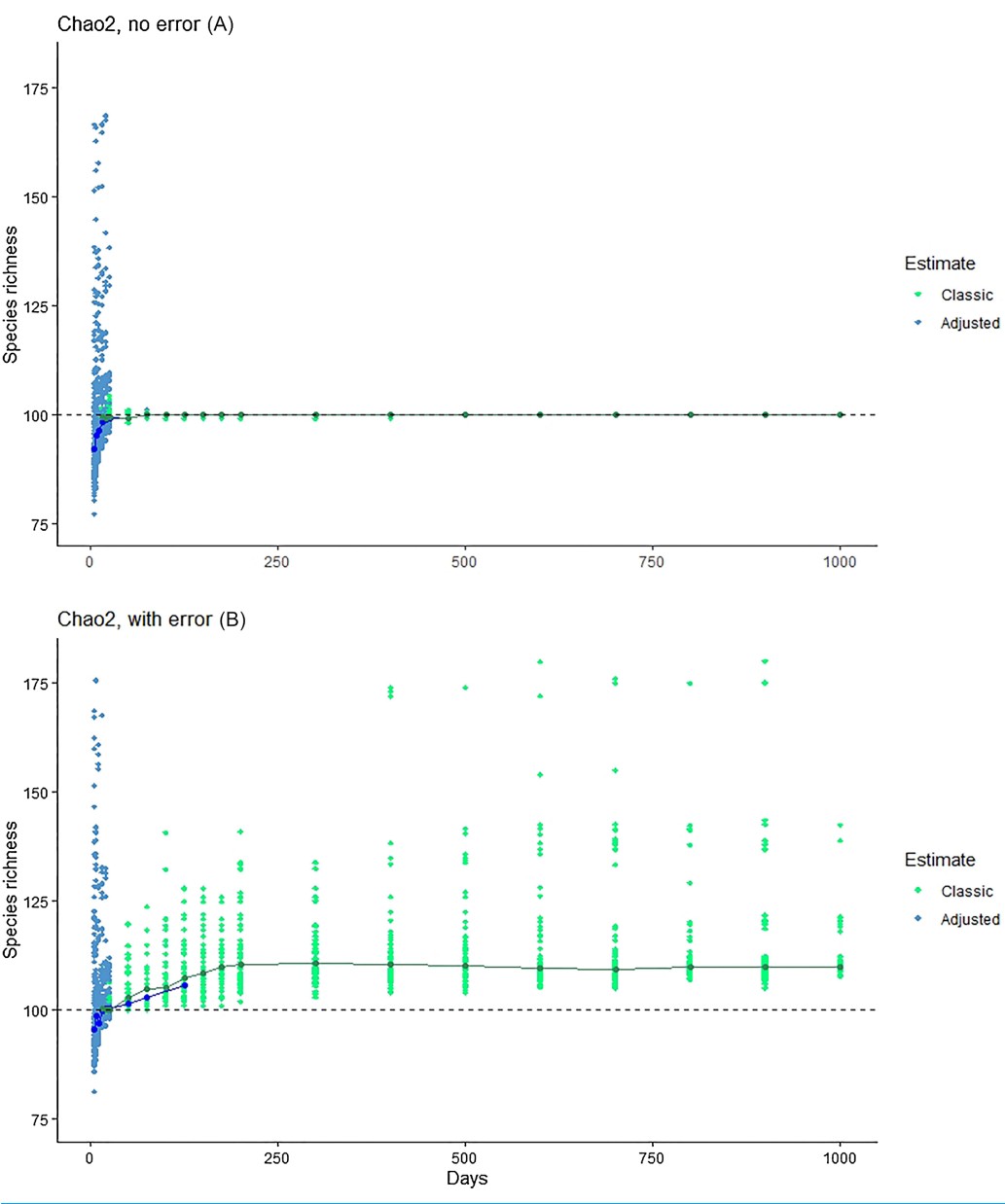

**Figure 4 Exploration of the estimation method of *Chiu & Chao (2016)*, which takes into account $Q_1$, $Q_2$, $Q_3$ and $Q_4$.** Note that, at larger sample sizes, the Chiu-Chao estimator (blue points) defaults to the Chao2 estimator (green points; *Chiu & Chao, 2016*). We provide (A) the results for Chao2 (no error) for purposes of comparison, and then the results from the new estimator in simulations without (middle) and with (B) errors included.

can be detected easily by comparison of observed species lists with authority lists (*Gueta & Carmel, 2016*); this sort of error is well-known to inflate species richness estimates in inventories (*Sousa-Baena, Garcia & Peterson, 2013*). However, these names may also be real names chosen by accident from controlled pick-lists or that refer to the taxon in question under an older taxonomic concept; such errors may be hard to detect owing to the fact that they are valid names, but just not represented at the site in question. Similar

contrasts in detectability of different error types have recently been documented for ecological niche modeling and species distribution modeling (*Simões & Peterson, 2018*). At a most fundamental level, the only means by which to detect such taxonomic complications and remove them or fix them prior to analysis is careful vetting of data records by taxonomic experts.

A further complication may arise from errors in geographic referencing of occurrences. This sort of error may arise from careless data transcription, or from inaccurate post hoc georeferencing steps (*Peterson et al., 2018*). The effect, however, can be that of creating an occasional report of a species from a site where it does not have populations, which can feed into the sort of problems that are examined in this paper by creating the appearance of rare species in the sample.

Finally, and particularly for the case of birds and a few other taxa for which species are well documented, a third class of problems regarding species names may arise. Specifically, rare visitors, often termed vagrants, are valid species names, and the species may genuinely be present at the site at some (rare) point in time (see Fig. 1). However, depending on the specific definition of the biota under consideration and that is the target of the inventory, these species may or may not be relevant. That is, detection and documentation of such species depends on continuous, intensive presence of observers or collectors, and also on the presence of the "experts" who will be experienced enough to detect and report such records, and whose records of such species will be believed and accepted. Such dependencies will easily create biases that may make certain sites appear richer in species, when in actuality they are richer only in high-level observers (*Dittmann & Lasley, 1992*). More generally, this point serves to indicate that biotic inventories need to be defined carefully in terms of a particular point or span of time and space.

The method presented by *Chiu & Chao (2016)* was developed for application to microbial molecular diversity data to account for inflation of singletons by sequencing errors, which would appear to be closely akin to problems created by identification errors in species inventories. This method estimates the true value of $Q_1$, based on $Q_2$, $Q_3$ and $Q_4$, and uses the adjusted value in asymptotic diversity estimates. This estimator, in our simulation-based assessments, underestimated true species numbers in the absence of error, but estimated the true species number closely when errors were introduced—as such, the Chiu–Chao estimator may offer a useful solution to the problems identified in this contribution for biodiversity inventory estimates.

## CONCLUSIONS

In this note, we point out and document a complication with application of the commonly used species inventory statistics, as biodiversity data sets grow to be large. The base observation is that fauna sizes are finite, but sampling effort can grow without limit, which shifts distributions of frequencies of observations of species towards larger and larger numbers—this phenomenon has the effect of reducing the numbers of relatively rare species that inform inventory statistics. Two processes are involved: (1) estimators depend on the frequencies of detection of the rarer species, which decline to nil in very large

datasets; and (2) erroneous reports come to dominate the estimation process because errors are rare and real species accumulate much larger numbers of observations, such that estimates can come to be based entirely on noise rather than on signal. The first point is a simple consequence of massive-scale sampling of finite biotas; the second, however, derives from the dependance of inventory statistics on information from rare species. Solutions to these problems must involve detailed cleaning and quality control of data, and careful definition of the relevant species pool that is under study. Exploration of new estimators that take into account species with greater numbers of records or that correct for biases in $Q_1$ (*Chiu & Chao, 2016*)—may provide solutions to these problems.

## ACKNOWLEDGEMENTS

We thank the University of Kansas Ecological Niche Modeling Group for their support and interest in the course of this project. We thank Jorge Soberón for a helpful review of the manuscript. We also thank Anne Chao for leadership in this field, and for willingness to provide comment and resources necessary for this project.

### Funding
The authors received no funding for this work.

### Competing Interests
The authors declare that they have no competing interests.

### Author Contributions

- Ali Khalighifar conceived and designed the experiments, performed the experiments, analyzed the data, authored or reviewed drafts of the paper, and approved the final draft.
- Laura Jiménez conceived and designed the experiments, performed the experiments, analyzed the data, prepared figures and/or tables, authored or reviewed drafts of the paper, and approved the final draft.
- Claudia Nuñez-Penichet conceived and designed the experiments, analyzed the data, prepared figures and/or tables, authored or reviewed drafts of the paper, and approved the final draft.
- Benedictus Freeman conceived and designed the experiments, prepared figures and/or tables, and approved the final draft.
- Kate Ingenloff conceived and designed the experiments, prepared figures and/or tables, and approved the final draft.
- Daniel Jiménez-García conceived and designed the experiments, performed the experiments, analyzed the data, prepared figures and/or tables, and approved the final draft.
- Town Peterson conceived and designed the experiments, analyzed the data, authored or reviewed drafts of the paper, and approved the final draft.

## Data Availability

Data is available at KU ScholarWorks [1253]

http://hdl.handle.net/1808/25686 (simulations) and

https://github.com/LauraJim/SpeciesRichness (analysis).

## Supplemental Information

Supplemental information for this article can be found online at http://dx.doi.org/10.7717/peerj.8872#supplemental-information.

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
