# Peer review of "Inventory statistics meet big data: complications for estimating numbers of species"

_PeerJ, doi:10.7717/peerj.8872_

## Round 0.1 · original submission · Major Revisions

The reviewers both agreed that this will make an important point in a rigorous way. Please pay special attention to the request for more information and for additional analyses. In responding to request for more analysis, please carefully consider whether they could add to the manuscript - and, if not, give a detailed explanation of why not.

Finally, I think the manuscript could benefit from a more detailed exploration of what could be done to account for these problems. You suggest that better curation can account for some issues - and a modified analysis (that tends to undercount species) for other issues - but is systematic undercounting really the answer. I think you miss an opportunity to give more guidance - should strays be ignored? Also, what role does curation and platform design play in reducing these errors? If mistaken identities are the problem, then wouldn't they be more common - unless the platform for reporting is well-curated? Also, considering the combination of mutliple data sources, how much could discrepancy in taxonomic concepts be the main factor underlying these problems?

·

Basic reporting

In this manuscript the authors draw attention to the effects of small errors in biodiversity databases on the species richness estimations generated by means the use of nonparametric estimators. The cited literature is up to date and the purpose is clear.

Experimental design

This study is in fact a cautionary note aimed to highlight the dramatic effects in these estimations when occurrence data are managed uncaringly, containing taxonomic or occurrence errors. Using simulated assemblages and introducing a 10% of erroneous rare species, the authors demonstrate that the appearance of these species causes overestimations when the number of replicates increases. The rationale and the provided results are clear and the message that is sent to the users is useful

Validity of the findings

The message of this study could be yet more useful if different numbers of “error” species are included in the analyses (e.g. 1%, 3%, 5% and 10%) Please consider that in the example provided in the figure 1 only 3% of the species are erroneous. Additionally, I also suggest that the results and the figures 3 and 4 should include the mean number of species estimated when these rare species are introduced, also providing the 95% confidence interval. This may contribute to better understand the real contribution of these erroneous occurrences in species richness estimations.

Additional comments

I consider that this manuscript can be a nice and useful piece of information for all people interested in managing the data provided by biodiversity databases. I would encourage the authors to extend your results.

·

Basic reporting

This paper explores the effects of very rare occurrences in large datasets on estimates of species richness through a set of simulations. As often-used estimates rely on the distribution of either abundances in a sample or frequencies of incidence throughout sample sets, the analysis is relevant in the field, for it provides a warning about a potential source of overestimations of richness. The authors have followed a sensible, classically empirical analytical path: let’s create an artificial assemblage, sample it as if it was real, and observe how critical parameters (singletons and doubletons in incidence-based estimates) vary with sample size, and then introduce very rare incidences and compare whether the estimates hold or not.
The paper is correctly written, well referenced and structured, and presents the findings in a clear way. However, the Methodology section needs some clarifications.
I have no issue on the design of the experiment, and find it elegant and conceptually reproducible. In fact, I have tried to reproduce some of it, using a different, independent set of tools, to confirm its findings, and was able to get to similar results but only under somewhat different conditions: namely, with uniform or normal distributions for the samples, or, if using lognormal distributions as described in the paper, only for much larger samples.
I believe that the reason I have not been able to fully verify the results may not lie in the original experiment, but, rather, on how it is explained in the paper and therefore my ability to fully grasp all the details. When designing my own simulations, I struggled to find some parameters I think were missing. If they were not necessary, then I did not understand some important point. In either case, the paper needs some clarification. Very likely these clarifications may enable much better reproducibility. Then, whether the results hold or not will be up to the readers repeating the experiment as intended by the authors.

Experimental design

The first, main issue is the definition of “samples”, “replicates”, and “days”. They seem to be used interchangeably but in a rather confusing way. For example, all figure footings refer to “sample days” while the main text refers to “samples”, but the context seems to indicate that sampling days and (part of the) samples are replicates. Throughout the paper there is an inconsistency abut what a sample is. My interpretation (after several reads and examination of figure 1) is that a “sample” in this context is a set of 5, 7, 10… inventories, taken on a “day” each; that each inventory contains a set of incidences of [up to] 100 species; that the contingency matrix that condenses the samplings can therefore be 5, 7, 10… columns wide, and that each of these matrices has been redrawn up to 100 times. It is paramount to describe the simulations in good detail, and that the figure footings are consistent with the main text.
The second important issue (or not?) is that the “sample size” seems to refer to the number of inventories in the sample, which is correct for incidence data. However, a simulation of incidence data requires another parameter: What are the sizes of the populations being sampled within the assemblage? The paper states that the assemblage is log-normally distributed and the distribution parameters are supplied, which enables us to calculate the relative sizes of each population (=species) and therefore the probability of occurrence in the inventory relative to other populations in the assemblage. Yet, whether a species is “sampled” (i.e., appears in an inventory) depends on that relative probability but also on the sample size: number of individuals in the inventory. By definition, a species occurs in an inventory if at least one single individual is sampled (n=1), and all species being sampled in just one inventory within a multiple (=multi-inventory) sampling contribute to Q1; but the parameters of the lognormal distribution, which are the average and standard distribution of ln(x), are given in the text (or so I believe) in relative terms, despite being presented as average abundance. In order to create the assemblage, the probabilities must be applied to a certain population size. For a 100-species assemblage following a lognor distribution with average abundance = 1.5 and a standard deviation = 2, a heap of species would be represented by a fractional individual, which is unrealistic. Thus, an overall abundance (assemblage size) should be given ensuring that the 100-species complete inventory includes at least one individual of the least-represented species.
When using such inventory sizes, simulations seem to show that Q1 and Q2 do not saturate as described in the paper but take longer, unless the underlying distribution is much more uniform.

Validity of the findings

See comments above.

Additional comments

Other lesser issues on the paper can be easily corrected, and are annotated in the PDF. For example, (line 82) the authors have explored four nonparametric estimates (not three as written). There is also a panel missing in figure 4 (footing says 3 and there are only 2).
My recommendation is that the paper will be publishable when authors clarify the details of their simulation enough for other researchers to be able to reproduce the results without second guesses. A good rewrite of the Methodology would be highly convenient. Indeed, supplying a sample matrix of the artificial assemblage and a worked-out example of the simulation would be quite useful, in addition to the already-supplied scripts. These alone do not seem enough to fully and reliably reproduce the reported results.

---

## Round 0.2 · accepted · Accept

You have addressed all the issues in the original reviews and I think that your paper makes a very important point. I hope that it helps step up the care with which people do these kinds of studies!

·

Basic reporting

In my former suggestions I recommended including different numbers of “error” species, but the authors reject this suggestion because it “will come to the same conclusion as those that we have already presented, and will not reveal any new properties”. This is probably true, although could improve understanding the linear or non-linear character of the relationship between the number of these errors and the overestimations. Be as it may be, as the own authors state, the results of this manuscript “demonstrate something that is entirely obvious”; i.e. that including rare species as consequence of erroneous occurrences or taxonomical identifications may provide erroneous estimations. Thus, taking into account that this study must be seen as a cautionary note, I accept the current version of the manuscript expecting that it improve the management by users of the biodiversity information with non-parametric estimators.

Experimental design

I consider that the provided methods are adequate

Validity of the findings

I consider that this manuscript can be a nice and useful piece of information for all people interested in managing the data provided by biodiversity databases

Additional comments

anyone